# Persuasive linguistic tricks in social media marketing communication—The memetic approach

**Krzysztof Stepaniuk** *, **Katarzyna Jarosz**

Department of Marketing and Tourism, Bialystok University of Technology, Kleosin, Poland

* k.stepaniuk@pb.edu.pl

**Data Availability Statement:** The data is available at the Mendeley Data repository at https://data.mendeley.com/datasets/2tcfrpw65n/2.

**Funding:** This research is supported by the Bialystok University of Technology and financed by

## Abstract

The paper reports the results of a study into the use of linguistic cues defined as Persuasive Linguistic Tricks (PLT) in social media (SM) marketing communication. It was assumed that the content shared on Social Networking Sites (SNS) could be perceived as specific sets of meanings (memeplexes), where a single component, also PLT, may function as their part. Following an original typology of PLT, created based on an emotional factor, the research focused on whether and how the number of positive, neutral and negative PLT used in Facebook posts impacted the behaviour of content recipients. These activities, including liking, commenting and sharing, are strictly connected with post spreading and range. The data analysis focused on 167 Facebook posts shared by five leading Polish travel agencies and 1911 responding comments. The quantitative content analysis method and Spearman's correlation tests were used. A relationship was observed between the number of emotionally positive and neutral PLT and the increase in the range of content with these PLT. The use of PLT by post recipients was also observed in their comments. This phenomenon is possibly related to the memetic nature of PLT. From the perspective of marketing messages, the obtained results contribute to and guide the textual content-building with a high spreading potential owing to the memetic capability of PLT. Further elaborations were made on the assumption for the evolutionary approach in social media content transfer and its processing.

## 1. Introduction

Social networking sites (SNS) are platforms enabling user interactions and information diffusion [1]. The latter was defined by Rogers [2] as a process of information spreading from a source to a recipient or groups of recipients through communication channels. Such spreading is based on specific relationships established between network members [3]. Enterprises in a marketing communication use the specificity of relations resulting from common interests, social affiliations, attitudes, and behaviours to target messages at specific recipients using a mix of text elements, graphical and/or multimedia content [4].

Language is the basic carrier of information [5] and the tool for the expression of meanings. It consists of semantic signals (expressed as written or spoken content), stimulating human

a subsidy provided by the Minister of Science and Higher Education, grant number WZ/WIZ-INZ/2/2019. The funders had no role in study design, data collection and analysis, decision to publish, or preparation of the manuscript. The authors did not receive any remuneration from Minister of Science and Higher Education and any other founders.

**Competing interests:** The authors have declared that no competing interests exist.

minds to build abstract and structured representations of presented elements in a sequence [6]. Language is a verbal representation of transferred meanings. It enables linguistic manipulation and impacts recipient actions [7]. Such manipulation often uses persuasive linguistic tricks (PLT), i.e., short phrases that contain hidden meanings. Messages with PLT aim to promote recipient behaviour desired by the source of the message [8] or, according to van Laer *et al.* [9], affective, cognitive and attitudinal reactions of recipients.

First, the study aimed to demonstrate that PLT could have a memetic character, as maintained by Vaneechoutte & Skoyles [10], because of a particular form, the ability to be transferred, and meaningfulness. The theoretical framework used to achieve this aim was based on the sender–recipient–effect model [11] and the SNS persuasion modelling proposed by Hong *et al.* [12]. The effect could be recognised as a specific recipient behaviour, expressed as commenting on the shared source content that directionally uses PLT.

Second, the study aimed to analyse the relationship between the PLT frequency of three emotional categories (positive, neutral and negative) [13] and the behaviour of the message recipients in SNS. According to Djukic [14], Wang and McCarthy [15], a strong correlation exists between the content range and behaviour of recipients, i.e., liking, posting, commenting and sharing the content [16, 17]. PLT present in the textual content could be one among different types of factors promoting such recipient behaviours.

The research has a utilitarian character as results can be incorporated into a process of the textual content creation in social networks marketing communication focused on increasing the content range and specific recipient behaviour connected with the further use of PLT as components of the primary transmission. This study of persuasive language tricks and their use in marketing communication in social media represents a new, interdisciplinary approach to linguistic persuasion from the evolutionary perspective.

The paper is organised as follows: Section 2 presents the theory; Section 3 introduces hypotheses and the aims of the work; Section 4 contains the data and methodology; Sections 5 lists the study results; and Section 6 gives the discussion. The article ends with a description of research limitations, a short description of possible future research and conclusions.

## 2. Literature overview

### 2.1. Memes and marketing communication

Memes [18, 19], like words, are information carriers, representing particular meanings. They are defined as "bits of behaviourally transmissible information" [15]. Consequently, memes are the direct factor shaping the perception of surroundings (entities, products, services, etc.) and enabling the expression of attitudes towards them [20] as well as specific behaviours due to their "capability of affecting individuals" [21].

Thus, marketing communication concerning a specific product, service, entity, phenomena or ideas can be described using sets of particular meanings (memes). First, such sets can provide a semantic description of a product, service, entity or phenomena as a whole. Second, components of those sets are present in different frequencies and, therefore, are subjects of the cultural selection processes [22]. According to Barnes [7], the language we speak shapes our reality, as the perception of the world and expression of our thoughts about it correlate with the words we use. Furthermore, the words we use influence recipients. According to the standard linguistic theory, meanings are attached to linguistic artefacts, and those meanings are materialised, transmitted and decoded by the speaker/writer and hearer/reader using the same mental machinery inside their brain [23]. Marketing communication in SM was analysed from the perspective of several aspects. In the case of politics and SM communication, De Paula *et al.* [24] showed the colossal role of symbolic acts (among others symbols, cultural

references which are not directly related to politics) in the creation and dissemination of the shared content as well as the development of the relationship between the content source and the recipient. Sheng [25] and Saboo *et al.* [26] investigated the entity–customer relation from the perspective of the online activeness of firms influencing comment-posting. Simultaneously, Kujur & Singh [27] described the vividness, interactivity, entertainment and information as characters of SNS marketing content directly influencing online consumer participation. All of the mentioned factors were crucial for creating recipient/user engagement using specific jargon, symbols, activities, etc. All of these elements were typical from the perspective of memes.

The theory of memes can significantly impact the creation and maintenance of customer relations in the contemporary and competitive market. Falát *et al.* [28] suggested that "the main goal of marketing communication is the transfer of the message between the sender and the recipient". The most apparent marketing communication role is to create and maintain customer relations and spread information about products or services. Simultaneously, marketing communication is connected with the creation of a semantic description of an image. Thus, due to the transfer of specified meanings, marketing communication is partly connected with shaping customer behaviour according to the wants of the message sender.

The memetic approach, despite the reservations expressed by Plotkin [29], is simple and compatible with the sender–channel–message–recipient–effect model [16, 30, 31]. It gives a wide range of possibilities for a qualitative and quantitative analysis of the content shared in social networks. Besides, it gives rise to the theoretical basis for scientific investigations concerning the present and later perceptions of the shared data.

Content posted on social networking sites is built from several sets of meanings expressed in different forms, which could be described as sets of memes—cultural semi-replicated information carriers. According to Heylighen & Chielens [32], such type of sets could be defined as memeplexes. Like genes, memes can change their frequency and range through their transferability within social groups "from one's memory to another's memory" [33]. Schlaile *et al.* [34] analysed the concept of a memeplex based on the ALS Ice Bucket Challenge diffusion process and stated that it was a recombination of meme-based and Internet-inspired behaviours: cold water challenge and neknomination. Thus, memes could be perceived as factors based on imitation [35] that shape or influence a wide spectrum of recipient activities (affective, cognitive and behavioural) among individuals and groups in real and digital life. From the perspective of content diffusion effectiveness, the range is a crucial aspect. Thomas and McDonogh [36] suggested that the range was strongly correlated with "how" and "what" source wanted to communicate and share. Communication, which is directly focused on activities generated in relation to transferred meanings, is also the basis for creating a web of relationships that can grow dynamically from the perspective of an increasing frequency of interactions between SNS users [37]. From the perspective of SNS, the description of recipient (user) activities was presented using the AIDAT model [13] and the COBRA model [14]. It refers to the creation of a hierarchical relationship between a cognitive stage (viewing content, which stimulates the recipients' attention), a behavioural stage with a moderate level of content consumption (liking, commenting, sharing), and a behavioural stage with a high level of content consumption (creating and sharing User-Generated Content based on the original message). All types of activities are derivative effects of the communication process, mainly including content formation, content transfer and processing, as well as its further spreading and range. From the perspective of the latter, the key issue is the frequency of meanings in SM communication.

All activities mentioned above mutually integrate virtual users based on Secondary Social Bonds (SSB)—weak relationships that lack a strong emotional context and are particular to group members clustered around common interests and related activities [38] and bear the

marks of homophily [39]. These activities relate to communication, commenting, exchanging and sharing information in forms of text, graphics and multimedia.

## 2.2. Linguistic tricks as an element of persuasive language

Hatim and Mason [40] suggested that advertisement content should be constructed as a specific continuum between informational and manipulative content and could be perceived as persuasive language. According to Labrador *et al.* [41], three-quarters of online advertisements include explicit persuasive phrases, which results from an attempt to draw the attention of recipients using affective words, informal expressions and metaphors.

Based on the Elaboration Likelihood Model of persuasion (ELM) [42], besides the central route of the persuasion process, mainly connected with deep analysis and processing of a message by a recipient, a peripheral route of the process exists, strictly associated with an emotional undertone of the message and its influence on the recipient's behavioural or attitudinal change.

McInerney [8] defined linguistic tricks as a meaning "that cannot be clearly discerned within the text". Linguistic tricks are hidden in the text and can be found only by an aware recipient. The use of such valuable resources inside the text aims to intensify the effect of marketing communication and make customers choose whether to take a particular course of action.

The intensity of linguistic tricks and the final result of their use vary and depend on their emotional character and frequency of occurrence. Xu *et al.* [10] suggested the typology of the content shared in social media and divided it into elements having a positive, neutral and negative character. Hong *et al.* [17] used Hovland's model of persuasion to describe several main elements acting in the process of content transmission: a persuader (SNS source of the message) [43]; persuading information(SNS message); the situation of persuasion (the whole content of an SNS webpage); and the recipient of the message (SNS user). Such an approach corresponds with the communication process theory according to Berlo [30] or other authors [31], [16].

Due to the properties mentioned above, positive, negative and neutral linguistic tricks in SNS marketing communication as persuasive language elements play a crucial role in the peripheral route of persuasion and could be perceived as a crucial factor in persuasive marketing communication in SNS.

For the sake of the current and future studies, a synthetic definition of PLT as an element of persuasive language was formulated, basing on the definition of persuasiveness according to Gilly *et al.* [44]: "the change in attitude and/or behavioural intention resulting from an interpersonal informational exchange". PLT are the semantic accidents with different emotional connotations, containing hidden meanings which might be specific descriptors of content they are representing and expressing as well as triggers stimulating specific activities or behaviours of recipients. Except for the holistic perspective on the perception of the phenomenon or the entity in social media described by the available content, PLT are a component of memeplexes and could also be perceived as elements influencing and modifying behaviours in the real world (mainly consumer behaviours) and the digital world (through user behaviour). Thus, their frequency of presence and emotional connotations are important.

Based on the analysis of linguistic and psychological literature [45, 46] and other resources mentioned in Table 1, a typology of PLT as elements of persuasive language according to the affective factor [10] was created.

## 3. Hypotheses and the aims of the work

Marsden [73] defined memetics as a paradigm of all expressed human behaviours, i.e., all of them are influenced by transferable cultural traits, reaching from the surrounding

**Table 1. Typology of persuasive linguistic tricks.**

| Persuasive Linguistic Tricks with positive emotional connotations |
|---|
| • "Agreement frames": I agree . . . and I want to add [47];<br>• "Amendment technique": information about a previous (higher and current) lower price of the product [48];<br>• "And that's not all": i.e., a special bonus is introduced [49];<br>• "Authority": people changing their decision-making process when confronted with the opinion of authority [50];<br>"Cueing": a product associated with something from the environment [51];<br>"Fun (anecdotes)": a product associated with the feeling of joy caused by a funny advertisement [51];<br>• "Group": becoming a part of a particular group by buying the product/service [52];<br>• "Elite (exclusivity)": becoming a member of an elite group by buying the advertised goods [53];<br>• "Hypnosis": using suggestions to create vivid pictures/emotions/sounds in a person's mind [54];<br>• "Independent point of view": the author of a message seems to have no intention of changing the reader's opinion [55];<br>• "Known": making an association with a similar product or with a person who uses the product/service and is well known, liked, and respected [52];<br>• "Majority": the text might give an impression that everybody thinks this way or that the majority shares the presented view [56];<br>• "Name": the name of the product should attract the attention of the customer [51];<br>• "Problems and solutions": presenting a common problem and showing the solutions [47];<br>• "Quotation": quotations about the content being trustworthy [56];<br>• "Rivalry": showing the advantage over other people, group, society, and companies which would be gained after buying the product [57];<br>• "Similarity": people like those who resemble them; using similarities as a wording strategy [58];<br>• "Uniqueness": products/services which are in limited edition are more desired [52];<br>• "VAKOG": content as a factor stimulating senses [59];<br>• "Visualisation": an imagined act of using the product [52];<br>• "Persuasive words": making the text more persuasive by using such words as you, discovery, easy, guarantee, safety, to save, health, love, new, proven, results, free [60];<br>• "Yet": this word suggests openness for change [60]. |
| **Persuasive Linguistic Tricks with neutral connotations** |
| • "Arguments": general arguments, arguments of the majority, the argument of a single case when something helped achieve the goal of a person [61];<br>• "Associations": words and phrases are associated with events, feelings and emotions [62];<br>• "Attention getter": attractive headline, subheadings, and short slogans drawing the attention of a reader [63];<br>• "Because": people are ready to do something on request when a person uses the word "because" [64];<br>• "But": this word allows the reader to consider only the last part of a sentence [46];<br>• "Comparing": comparing and contrasting objects allows consumers to make one of them easier to choose [47];<br>• "Emotion": arousing emotions in a person is a key factor in persuading somebody to make a particular decision [52];<br>• "Generalisation": may be linked to superstitions and stereotypes [65];<br>• "Imperative": the persuasive power of orders [46];<br>• "Metaphor": a figurative expression that easily creates a mental picture and shapes one's vision of reality [66];<br>• "Passive voice": suggestion that the course of events is or was unavoidable [67];<br>• "Precision": details make the decision-making process easier [52];<br>• "Presupposition": the presence of presupposition(s) in an utterance suggests that the statement is true and acceptable [68];<br>• "Redefinition": Basu [60] defined it as a pattern: "The problem is not about X, but about Y, and this means that. . .";<br>• "Repetition": a repeated statement is easier to understand and remember [69];<br>• "Slogans": shortened, generalised, and simplified expressions, easy to notice and remember; they seem to be true [46];<br>• "Story": the recipient's identification with the product [46];<br>• "String of nouns": many nouns in a sentence can bind together several ideas, events and opinions [70];<br>• "Superlative": the message might seem to be emotional. It can be achieved either through the use of superlative, e.g., the best, or through the use of some prefixes, e.g., super-rich [53];<br>• "The truth effect": information is not perceived as inaccurate, so it is treated as verified and true [71];<br>• "Vicious circle": a statement that aims to define a term and repeat this term in the definition [72]. |
| **Persuasive Linguistic Tricks with negative emotional connotations** |
| • "Cognitive dissonance": the creation of a need to change the element which disturbs internal coherence [55];<br>• "Limited choice": limitation of choice to a few alternatives [52];<br>• "Pain vs. pleasure": buying the product/service to avoid something unpleasant [52]. |

socio-cultural environment. Thus, every topic, i.e., phenomena, entity, service, product, etc., could be perceived and described as "sets of interlocking memes" [74], or memeplexes, i.e., meanings present in textual or visual content. From the perspective of the communication process, such a memeplex determines the perception of the topic represented by the source of content. Also, it could contain PLT related specifically to the topic as it is desired by the source of content. Thus, direct meanings and the PLT are present in specific frequencies forming a holistic message.

On the other hand, recipients consuming the source content also formulate their representations of the topic. Simultaneously, recipients acquire and accept or reject a part of transferred meanings and, through this process (similar to cultural selection), change the original frequencies of particular meanings (memes) and PLT and use the accepted meanings (and PLT) to express their perception further. Formation, transfer, exchange of memeplexes and possible effects of such processes are basic components of communication and simultaneously are the characters of WEB 2.0/3.0 technology where users and their perception and expression of the surrounding reality are crucial in the context of information and experience exchange. It mainly depends on user preferences and experiences, i.e., customer experiences. Also, it is directly connected with the phenomenon of collective intelligence [75]. In the case of online communities, Mačiulienė & Skaržauskienė [76] recognised and described three main dimensions of the collective intelligence in online groups: capacity (user activities and interactions), emergence (the ability to self-organise communitarians, based mainly on motivation), and social maturity (including the intellectual influence of a user group on the remaining society in the context of new knowledge/new attitudes/behaviour formation and further dispersion).

The presented research is located in the capacity dimension framework, defined as actions of individual users resulting in vast interactions and development as well as a spread of new knowledge and competencies—ideas, points of view [76, 77]—in the context of "observing the actions of others" and "decisions about activity". The communication theory is also crucial considering that all user activities are manners of the communication process and content dissemination is its effect. Thus, from the perspective of multiple transfer acts and memeplex processing, the collective intelligence concept could be a key research issue on content decomposition, content management and the impact on consumer behaviour.

The added emotional factor concerning objects/phenomena/products or services may result in their improper perception and a limited content range. Atadil *et al.* [78] maintained that the meanings expressed by individuals in relation to objects determine behaviour towards them. The frequency of occurrence and emotions attached to PLT in an SNS marketing message may contribute to their memetic transfer and further use by recipients as well as the content range. Simultaneously, Aronson *et al.* [79] suggested that emotions made advertisements effective due to the ability to capture the attention of recipients. All the above-mentioned activities of SNS users are related to content virality—a multi-level phenomenon focusing on several aspects, including, i.e., the issues of content spreading, the emotional component(s) of the content, and the emotional reactions of the audience and/or changes in their attitudes and behaviour (Strapparava *et al.*, 2011) [80].

Thus, emotions and linguistic tricks used in a marketing message can contribute to stimulating the behavioural activities of recipients and increase the range of shared content. The frequency of occurrence could be the crucial factor for the continuous duration and use of PLT. According to DeLosh & McDaniel [81], the frequent use of a word or phrase in the source content increases the probability for the further and more frequent use of the content by recipients. Thus:

H1: In a holistic approach to persuasive SNS communication, the greater is the number of PLT used in memeplexes, the more frequent is the subsequent use of the message by recipients.

Based on Méndez-Bértolo *et al.* [82], the emotional undertone is also significant. The authors stated that negative content determined faster recognition and processing of words. However, in the communication process, a positive emotional factor is also relevant. Gatti *et al.* [83] described the role of the speaker's accurate wording in connection with the specific/ desired emotional tone and the impact of both on the elicitation of similar affective attitudes in recipients of the presented content. Based on political speech analysis, Guerini *et al.* [84] showed that despite the declared positive attitudes to the speaker, recipients still tended to react emotionally to signals of persuasive communication. These results suggest that the emotional overtone of the message is taken over by the audience. However, Ferrara & Yang [85] and Naveed *et al.* [86] showed that in Twitter, negative content tended to spread faster, but its range was smaller compared to positive content mainly due to positive bias (the tendency of recipients to share and like positive content). Thus, the 2nd hypothesis was formulated as follows:

H2: In the holistic approach, i.e., concerning the whole volume of content shared by an entity in the communication process, the presence of persuasive language expressed as PLT with a positive emotional undertone triggers more recipient behavioural than the negative and neutral messages.

## 4. Data and methodology

Five leading Polish travel agencies were surveyed (all of them had active Facebook profiles and were top travel agencies in 2014) [87]. For the research we had not consult with Ethics or Data Protection Committee. In Poland, such consents are not required. We used only world-wide available text content shared by five leading Polish Travel Agencies as well as comments posted by users. Due to their marketing properties the posts as well as comments could not threaten personal privacy or damage the reputation of content authors and commenting users.: (1) Itaka, (2) Rainbow Tours, (3) TUI Poland, (4) Wezyr Holidays, and (5) Grecos. The shared textual content was searched for the presence of PLT—based on the typology showed in Table 1 —and the frequency of positive (variable $c_I$), neutral (variable $c_{II}$), and negative (variable $c_{III}$) PLT.

Simultaneously, to demonstrate the possible memetic character of PLT, the semantic content was analysed, focusing on 1911 comments posted by Facebook users in response to posts by travel agencies. The analysed comments were posted in the summer period, i.e., July–August 2015 (1104), and in the autumn and winter period, i.e., December–February 2015 (807). The researchers collected the comments posted in response to a specific shared item and analysed them in detail. One researcher noted the presence and the number of PLT found in each post. The recorded PLT strictly corresponded to the patterns indicated in the typology (Table 1). PLT were summarised and saved in a spreadsheet according to positive, neutral and negative emotional overtones.

Then, the content of the comments was checked. The presence of identical elements was also noted. According to assumptions of memetics, where imitation plays a major role in the transmission of content, such a result indicates the probable memetic nature of the transfer between the sender and the recipient. This can be the basis for a scientific theory concerning a memetic shaping of the reality perception (including products, brands and companies) and behaviour in SNS.

Aiming to determine a possible impact of emotive PLT on the post range, the assumptions of the Facebook Interactivity Index (FII) were used where each individual behaviour of social network users has a different weight. According to FII, one "like" is weighted as 1, a comment —as 4, and a share—as 16 [11]. Stepaniuk [88] measured the potential of content spread

among users based on the sum of weights assigned to the total number of likes, comments and shares for a single post using the formula $m\_a\_i = n\_li+4^*n\_co+16^*n\_sh$, where: $m\_a\_i$—the potential of the content range; $n\_li$—total number of likes; $n\_co$—total number of comments; $n\_sh$—total number of shares. It should be noted that in the formula, the number of likes does not refer to the emotional variants "like it" button, but to their total number, as having a direct impact on the spread and range of content. The Shapiro-Wilk test was used to check the presence or absence of the normal distribution of all analysed variables. The Kruskal-Wallis test was used to verify differences in the quantity of emitted emotive PLT by the analysed travel agencies [89]. Nonparametric correlation tests that are widely used in social networking exploratory data analysis [90] were used to assess the impact of the seasonality on the PLT frequency in published posts. The same method was used to analyse the relationship between the number of positive, negative and neutral PLT, the behaviour of social media users, and the content range, as well as for determining the potentially memetic character of PLT.

The data collection process and research were conducted between March 2016 and October 2017. The statistical analysis was performed using STATISTICA 13.3.

## 5. Findings

In the summer season, the analysed travel agencies publicised 72 posts (variable n_p_s). They were pictures with text (60), hyperlinks (10), hyperlinks with text (1), and short videos with text (1). The presence of PLT was noticed in 63 of them (Itaka—22; Grecos—15; Rainbow—15; Wezyr—6; TUI—5), which is 87.5% of all posts (Table 2). The autumn–winter season (variable n_p_w) saw 95 posts (TUI—22; Rainbow—17; Wezyr—14; Grecos—13). Most of them were pictures with text (n = 76), hyperlinks (n = 11), short videos with text (n = 7), and hyperlinks with text (n = 1). PLT were present in 87 of the posts, which is 91.6% (Table 1). None of the analysed variables had normal distribution, except for several summer–season posts (variable n_p_s; W = 0.93; p = 0.60) and some posts of the autumn–winter season (variable n_p_w; W = 0.87; p = 0.26).

The analysed content contained 556 PLT varying in their emotional tone. 238 PLT (variable c_I; $\bar{x} = 1.58$; $SD = 1.31$) had a positive emotional connotation, 289 (variable c_II; $\bar{x} = 1.92$; $SD = 1.53$) were neutral, and 29 (variable c_III; $\bar{x} = 0.19$; $SD = 0.44$) had a negative association. Each travel agency as a message source used a different number of PLT varying in their emotive character (Table 2).

The results of the t-test (c_I (t = -3.49; p = 0.0008), c_II (t = 0.088; p = 0.89), c_III (t = -2.003; p = 0.049)) and the Kruskal-Wallis (c_I (H = 20.38; p = 0.004), c_II (H = 16.4;

**Table 2. Number of PLT with different emotional character in the content shared by the analysed travel agencies.**

| | Summer season | | | | | Winter season | | | | |
|---|---|---|---|---|---|---|---|---|---|---|
| **Category I (c_I) (number of linguistic phrases with positive emotional-memetic connotations)** | n = 109 | | | | | n = 129 | | | | |
| | (1) | (2) | (3) | (4) | (5) | (1) | (2) | (3) | (4) | (5) |
| | 45 | 13 | 9 | 2 | 40 | 16 | 19 | 50 | 18 | 26 |
| **Category II (c_II) (number of linguistic phrases with neutral emotional-memetic connotations)** | N = 143 | | | | | N = 146 | | | | |
| | (1) | (2) | (3) | (4) | (5) | (1) | (2) | (3) | (4) | (5) |
| | 65 | 26 | 6 | 4 | 42 | 28 | 34 | 46 | 8 | 30 |
| **Category III (c_III) (number of linguistic phrases with negative emotional-memetic connotations)** | N = 11 | | | | | N = 18 | | | | |
| | (1) | (2) | (3) | (4) | (5) | (1) | (2) | (3) | (4) | (5) |
| | 2 | 3 | 1 | 1 | 4 | 4 | 0 | 8 | 4 | 2 |

Note. (1) Itaka, (2) Rainbow Tours, (3) TUI Poland, (4) Wezyr Holidays, (5) Grecos.

p = 0.0025), c_III (H = 3.24; p = 0.5)) comparison of ranks indicate the existence of statistically significant differences among the mean values of the number of the tricks among the travel agencies and depending on the season.

For summer, in terms of positive phrases (c_I), the multiple comparison tests showed that statistically important differences existed among the mean values of PLT publicised by (1) and (4) (R = 2.9; p = 0.03), (2) and (5) (R = 3.29; p = 0,009). The analysis (c_II) into neutral PLT revealed the differences in average values among the messages of travel agencies: (1) and (4) (R = 3.31; p = 0.009). In the analysis of the negatively associated PLT (c_III), no statistical differences in the average values were found.

In winter, statistically significant differences concerning the average content of the PLT of neutral character (c_II) applied to the posts of: (1) and (4) (R = 3.03; p = 0.02), (2) and (4) (R = 2.93; p = 0.03); (3) and (4) (R = 2,82; p = 0.047); (3) and (5) (R = 3.6; p = 0.003). For the positive (c_I) and negative (c_III) transfer, no statistically significant differences were found. The content shared by the analysed travel agencies (including LT) was received differently by recipients (Table 3).

## 5.1. The number of PLT and behavioural activities of recipients in SNS

The analysis of the content containing PLT (n = 63) shared in the summer season proved a link between the number of the positive PLT (c_I) with the number of likes (n_li; ρ = 0.45; p < 0.0001), the number of comments (n_co; ρ = 0.29; p < 0.02), and the number of shares (n_sh; ρ = 0.36; p < 0.003). For neutral PLT (c_II), the occurrence of statistically important correlation with the number of likes (n_li, ρ = 0.3; p < 0.01) and the number of shares (n_sh, ρ = 0.4; p < 0.001) was observed.

In the winter season (n = 87), the statistically important correlation was found only between the number of neutral tricks (c_II) and the number of comments (n_co, ρ = 0.22; p < 0.03), and the number of shares (n_sh, ρ = 0.32; p < 0.002). Details are presented in Table 4.

## 5.2. The number of emotional PLT and the post range

For the summer season, a positive, statistically important, weak correlation was proven between the number of emotive PLT and the range of the post, calculated based on the sum of weights assigned to the total number of likes, comments and shares for a single post (m_a_i variable). It was found both for positive (c_I) ($\rho_S = 0.41$; p < 0.05) and neutral PLT (c_II) ($\rho_S = 0.35$; p < 0.05). Although the relationship was weak, the greater was the number of positive and neutral PLT included in a post, the wider was the range of the post. For the winter season, similar positive and weak correlation exists only for neutral PLT (c_II) ($\rho_S = 0.27$; p < 0.05). Details are presented in Table 5.

**Table 3. Descriptive statistics concerning recipient behaviour of the content shared by travel agencies on Facebook.**

|  | n_li | | n_co | | n_sh | |
|---|---|---|---|---|---|---|
|  | s_s | w_s | s_s | w_s | s_s | w_s |
| Σ | 10529 | 8040 | 1104 | 807 | 1076 | 1126 |
| Average | 167.1 | 92.41 | 17.52 | 9.27 | 17.08 | 12.94 |
| SD | 232.3 | 133.6 | 232.3 | 14.48 | 33.3 | 29.57 |
| Min | 0 | 0 | 0 | 0 | 0 | 0 |
| Max | 1000 | 567 | 105 | 66 | 147 | 235 |

Note. s_s–summer season; w_s–winter season; n_li–no. of likes; n_co–no. of comments; n_sh–no. of shares.

**Table 4. Spearman's correlation coefficient values for the number of emotive PLT and their influence on the behaviour of SNS users (\*—significant correlation; p<0.05).**

| Correlation analysed | Summer season | | | | Winter season | | | |
|---|---|---|---|---|---|---|---|---|
| | N | ρ | t(N-2) | p | N | ρ | t(N-2) | p |
| c_I & n_li | 63 | 0.46* | 4 | 0.00* | 87 | -0.12 | -1.13 | 0.26 |
| c_I & n_co | 63 | 0.29* | 2.35 | 0.02* | 87 | -0.21 | -1.96 | 0.05 |
| c_I & n_sh | 63 | 0.36* | 3.04 | 0.00* | 87 | -0.21 | -1.97 | 0.05 |
| c_II & n_li | 63 | 0.31* | 2.55 | 0.01* | 87 | 0.23* | 2.16 | 0.03* |
| c_II & n_co | 63 | 0.21 | 1.69 | 0.10 | 87 | 0.18 | 1.69 | 0.09 |
| c_II & n_sh | 63 | 0.40* | 3.43 | 0.00* | 87 | 0.32* | 3.15 | 0.00* |
| c_III & n_li | 63 | -0.01 | -0.07 | 0.94 | 87 | -0.12 | -1.07 | 0.29 |
| c_III & n_co | 63 | 0.03 | 0.24 | 0.81 | 87 | -0.13 | -1.24 | 0.22 |
| c_III & n_sh | 63 | 0.08 | 0.63 | 0.53 | 87 | -0.19 | -1.81 | 0.07 |

Note. c_I–number of linguistic phrases with positive emotional-memetic connotations; c_II–number of linguistic phrases with neutral emotional-memetic connotations; c_III–number of linguistic phrases with negative emotional-memetic connotations; n_li–no. of likes; n_co–no. of comments; n_sh–no. of shares.

## 5.3. Emotional PLT and the memetic transfer

72 posts publicised by the researched travel agencies in the summer season received 1104 comments. The semantic content analysis showed that phrases containing PLT were found in 395 (i.e., 36%) of them (positive—353 (32.2%), neutral—40 (3.6%), and negative—2 (0.2%)).

For the summer season, there was a positive, weak relationship between emitting positive PLT and their presence in recipient comments (c_I_u$_s$, $\rho_S$ = 0.39; p < 0.05). Probably, it suggests the memetic character of the PLT transfer. The relationship was statistically insignificant for the presence in comments of neutral (c_II_u$_s$, $\rho_S$ = 0.05; p < 0.05) and negative (c_III_u$_s$, $\rho_S$ = 0.1; p < 0.05) tricks,. Details are presented in Table 6.

For the winter season, the researched travel agencies shared 95 different posts on Facebook. In response, the social network users posted 807 comments. 127 (i.e., 16%) had identical phrases as in the source message (c_I_uw—positive: 12% and c_III_u$_s$—negative: 4%). No statistically significant relationship was observed between emitting PLT and their presence in recipient comments.

In terms of holistic results, representing each travel agency separately but without the division into summer and winter seasons, the moderate relationship (c_I, $\rho_S$ = 0.58; p < 0.05) was found in the case of Grecos Travel Agency, and the weak relationship was present in the case of Rainbow Tours (c_I, $\rho_S$ = 0.26; p < 0.05).

**Table 5. Spearman's correlation coefficient values for the number of emotive PLT and their influence on the post range (\*—significant correlation; p<0.05).**

| Correlation analysed | Summer season | Winter season |
|---|---|---|
| c_I & m_a_i | 0.41* | -0.20 |
| c_II & m_a_i | 0.36* | 0.27* |
| c_III & m_a_i | 0.03 | -0.19 |

Note. m_a_i—the potential of the content range; c_I–number of linguistic phrases with positive emotional-memetic connotations; c_II–number of linguistic phrases with neutral emotional-memetic connotations; c_III–number of linguistic phrases with negative emotional-memetic connotations.

**Table 6. Spearman correlation coefficient values for the number of emotive PLT and their presence in users comments (\*—significant correlation; p<0.05).**

| Correlation analysed | Summer season | Winter season |
|---|---|---|
| c_I & c_I_u$_s$ | 0.39* | -0.15 |
| c_II & c_II_u$_s$ | 0.05 | 0.04 |
| c_III & c_III_u$_s$ | 0.16 | 0.00 |

Note. c_I–number of linguistic phrases with positive emotional-memetic connotations; c_II–number of linguistic phrases with neutral emotional-memetic connotations; c_III–number of linguistic phrases with negative emotional-memetic connotations; c_I_us–number of linguistic phrases with positive emotional-memetic connotations present in recipient comments in summer season; c_II_us–number of linguistic phrases with neutral emotional-memetic connotations present in recipient comments in summer season; c_III_us–number of linguistic phrases with negative emotional-memetic connotations present in recipient comments in summer season.

## 6. Discussion

The research results showed that the analysed travel agencies used a varied number of emotional PLT as an element of persuasive language in SNS marketing communication. Depending on the season, a smaller or larger number of PLT were used, which suggests that the autumn–winter period requires greater efforts in terms of SNS marketing communication. Using the evolutionary approach, it can be perceived as an adaptive strategy to the dynamically changing and competitive market.

Regardless of the season, the majority of PLT used in marketing communication were phrases with a neutral emotional undertone (c_II). The content with neutral PLT affects the behaviour of SNS recipients (perceived from the perspective of the number of likes, comments and shares). Consequently, the organic range of the post was increased. A similar but weak relationship was observed for positive tricks (c_I). No such correlation was observed for elements with a negative undertone (c_III), which were the least frequently represented in the sample.

Lim [91] stated a rational marketing message was more effective than emotional (emotionally neutral) from the point of view of shaping positive customer attitudes towards products or services.

According to Schiffman and Kanuk [92], rational messages, i.e., without an emotional undertone, were more acceptable and more influential for adult and well-educated customers from the perspective of purchasing decisions.

Thus, a positive emotional factor (c_I) connected with persuasive language could be viewed as a trigger for customer interest. A neutral message related to information about products or services (c_II) may be associated with building positive attitudes directly towards these products or services.

The obtained results suggest that the 1st hypothesis is most likely true. Although the number of emotionally positive PLT has a positive correlation with their use in recipient comments posted in response to the source message, the application of numerous PLT, tripartite in emotional character, could be perceived as modifiers of memeplexes of the entity/brand/product in SNS as well as a factor that creates and enhances their holistic representation and expression. At the same time, these sub-pools can be used as a tool for expressing the identity of a specific issue.

The 2nd hypothesis is also most likely true. Depending on the season, positive correlations between the number of positive and neutral PLT and the content range were observed. This is probably another confirmation of the truth of Pollyanna's hypothesis [93]—the tendency of

humans to favour and use positive rather than neutral or negative items in communication processes.

No relationship was observed between the number of negative PLT and the content range resulting from users; behaviour, mainly due to a diverse number of analysed PLT within emotive groups. However, full verification of both hypotheses requires further, more detailed research.

The way the recipient reacts to the message containing linguistic tricks is different each time and depends on the "phenotype" of the recipient defined as a set of socio-economic and cultural characteristics [15] based on Ridley [94]. It must also be stated that marketing communication is usually profiled and aimed at a particular group of recipients. A target group can be described in terms of their attitude towards certain values, behaviour, etc. The criteria for the classification of target groups can include linguistic tricks and susceptibility to them. A similar view on a brand personality is presented by Ivens & Valta [95], suggesting that its perception depends on the subjective evaluation of the message content and a way of interpretations by recipients.

In particular, it should be highlighted that the obtained correlation coefficients have low values for all analyzed variables. This may be due to the limited size of the research sample. It is possible that for larger samples the values of the correlation coefficients would be different. This may be explained by the fact that every post is not only text but also graphics and/or multimedia. These are factors that have a holistic impact on the recipients, possibly reducing the isolated impact of the text message alone. It should also be emphasized that every recipient may perceive and react to the shared content in a different way. This may also be the reason for low values of correlation coefficients.

PLT could also be used as elements of specific jargon—special expressions, typical for groups of users with specific interests and/or activities—to develop communities connected with brands, products, activities and/or services. According to Muniz & O'Guinn [96], such a type of practices might be perceived as an element building the sense of collectiveness and loyalty of groups. Simultaneously, a proper set of PLT could be a factor that somehow connects the brand/product, etc., with an online consumer community, and, thus, a factor distinguishing it from others through the use of such specific phrases.

PLT are also a type of information carriers with an impact on the recipients' behaviour in SNS. Their presence in user comments can be perceived as memes because they have been processed, adopted and manifested later. This process was observed only in summer, in the context of positive PLT. As similarly created patterns of behaviour can spread in a group of people thanks to imitation processes, they can be defined as memes. They are expansive (i.e., if they guarantee to obtain a subjective evolutionary advantage for their senders and users), and the frequency of their occurrence and spreading can be high.

## 7. Limitations of the study and future research

The presented results concern general issues related to isolation and the analysis of the influence made by three main emotional groups of PLT. This is one of the main limitations of the study. Moreover, due to the low representation of negative PLT, it is not possible to fully determine their actual role in disseminating the marketing message in the SNS. This justifies the need for further, more detailed research. At the same time, the investigated issues were related to marketing communication on one social media platform. The textual content was, therefore, relatively short. It is likely that some new, different relationships can be shown by analysing longer fragments of text, other SNS platforms or communication channels, as well as the spoken language. It may be important to examine the role of the syntax of marketing messages,

and its modification in relation to the presence of PLT indicated in the proposed typology. Similar research in relation to marketing messages on Facebook was presented by Atalay *et al.* [97]. The authors showed that the modification of the syntax of marketing messages directly influenced the attitudes declared by the recipients of the message as well as Facebook click-through rate (CTR). Further research should also focus on systematising the typology of persuasive language in the context of PLT used for SNS marketing communication. Moreover, researchers should undertake qualitative and quantitative analysis of the influence made by PLT on recipient activities, the range of the content and their memetic character in specific recipient groups. This will create a scientific basis for building the theory of memetic content transfer and their processing and management (in SNS but not only). Furthermore, it would be interesting to consider an evolutionary approach in future research. Assuming that PLT are parts of memeplexes, they probably spread according to the rules of evolution. Thus, some PLT could have more spreading potential than others. Some of them could have more spreading potential than others. Therefore, further research could integrate the evolutionary and management perspectives into one theoretical and methodological approach and, thus, open a new interdisciplinary research field.

Although most English phrases have equivalents in other languages, due to the specificity of national languages, the current typology, based on English-language scientific literature, may not be complete. In order to avoid the possible omission of certain phrases specific to national languages, the typology used should be expanded in further research.

## 8. Conclusions

Persuasive linguistic tricks with an emotional undertone in the content shared on social media can influence the activities of recipients and the organic range. Furthermore, in the case of positive PLT, the transfer between the source of the message and the recipient probably has a memetic character. Differently than in the case of negative content, a positive, emotional undertone of persuasive linguistic tricks, as a consequence of the real message hidden between the lines, is also a factor having an impact on the perception of the shared content. The behavioural reactions and activities of users vary according to topicality and scope of the shared content, which is supported by the results of analysis concerning the issue of seasonality, PLT and user behaviour. The summer season implies a positive relationship between the number of positive messages and user activities. From the perspective of the winter season, a similar relationship was observed in relation to neutral PLT.

The reflections presented above have a utilitarian character as they can be incorporated into social media content management or in the field of managing any published linguistic content. When used in marketing communication, small language components that have an element of persuasive language, such as PLT, may possibly influence the spread of shared content. In this case, which is supported by obtained results, the crucial issue is the number of PLT, their emotional character and the time of use.

## Author Contributions

**Conceptualization:** Krzysztof Stepaniuk.

**Data curation:** Krzysztof Stepaniuk, Katarzyna Jarosz.

**Formal analysis:** Krzysztof Stepaniuk, Katarzyna Jarosz.

**Funding acquisition:** Krzysztof Stepaniuk.

**Investigation:** Krzysztof Stepaniuk, Katarzyna Jarosz.

**Methodology:** Krzysztof Stepaniuk.

**Resources:** Krzysztof Stepaniuk.

**Supervision:** Krzysztof Stepaniuk.

**Validation:** Krzysztof Stepaniuk.

**Visualization:** Krzysztof Stepaniuk.

**Writing – original draft:** Krzysztof Stepaniuk, Katarzyna Jarosz.

**Writing – review & editing:** Krzysztof Stepaniuk.

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
