## [Decision Letter · Decision Letter 0]

2 Mar 2021

PONE-D-21-02199

Persuasive Linguistic Tricks in Social Media Marketing Communication - Memetic Approach

PLOS ONE

Dear Dr. Stepaniuk,

Thank you for submitting your manuscript to PLOS ONE. After careful consideration, we feel that it has merit but does not fully meet PLOS ONE’s publication criteria as it currently stands. Therefore, we invite you to submit a revised version of the manuscript that addresses the points raised during the review process.

Please, find below the detailed reviews that you are asked to carefully consider while preparing your revision.

We look forward to receiving your revised manuscript.

Kind regards,

Stefano Cresci

Academic Editor

PLOS ONE

Journal Requirements:

2. PLOS ONE has specific requirements for studies using personal data from third-party sources, including social media, blogs, other internet sources, and phone companies (https://journals.plos.org/plosone/s/submission-guidelines#loc-personal-data-from-third-party-sources). These requirements include confirming data are collected and used in accordance with the company or website’s Terms and Conditions, obtaining appropriate ethics or data protection body review, and ensuring appropriate consent from individuals whose data are used in research. In this case, please ensure that your Ethics statement is in compliance with guidelines, and that you have complied with the company's (i.e., Facebook's) Terms and Conditions, with appropriate permissions.

*In line with PLOS' guidelines on ensuring that methods and reagents are described in sufficient detail for another researcher to reproduce the experiments described (https://journals.plos.org/plosone/s/criteria-for-publication#loc-3), please attach the posts and comments included in the analyses as an "Other" file.

3. You indicated that ethical approval was not sought for your study. Could you please provide further details on why your study is exempt from the need for approval and confirmation from your institutional review board or research ethics committee (e.g., in the form of a letter or email correspondence) that ethics review was not necessary for this study? Please include a copy of the correspondence as an ""Other"" file.

 "NO"

Additional Editor Comments:

Thank you for giving us the chance to assess this manuscript.

Both reviewers found merit in the work, however, at the same time they also highlighted several areas requiring additional work. In particular, multiple sections require improved/extended explanations and additional details. Moreover, a better framing of the scientific contributions of this work, its objectives and its position with respect to the existing literature are also needed.

Finally, all reviewers required several thorough rounds of proofreading in order to improve the clarity and readability of the paper.

Pay close attention to the reviewers' comments and address their concerns before resubmitting the manuscript.

Please, also note that one reviewer provided feedback in an attached file rather than inline.

Reviewers' comments:

Reviewer's Responses to Questions

**Comments to the Author**

1. Is the manuscript technically sound, and do the data support the conclusions?

Reviewer #1: Partly

Reviewer #2: Partly

2. Has the statistical analysis been performed appropriately and rigorously? 

Reviewer #1: Yes

Reviewer #2: No

3. Have the authors made all data underlying the findings in their manuscript fully available?

Reviewer #1: Yes

Reviewer #2: No

4. Is the manuscript presented in an intelligible fashion and written in standard English?

Reviewer #1: No

Reviewer #2: No

5. Review Comments to the Author

Reviewer #1: __Full text of the review is submitted in the attached file__

Introduction

• A concise introduction is required to enable the reader’s understanding of the research problem.

• Expand to emphasize the problem leading to a clear set of research questions and objectives the research addresses.

Reviewer #2: The paper reports the results of a study devoted to investigate the use of linguistic cues in social media marketing communication with a specific focus on how they have an impact on the persuasion of social media users. The authors assume a memetic approach by assuming that the considered linguistic cues (named Persuasive Linguistic Tricks, PLTs) function as memes. More specifically, they devised a methodology that is articulated into the following steps:

- the authors collected a list of PLTs taken from the international literature (mostly in English) and classified the considered PLTs according to their polarity, i.e. positive, negative and neutral;

- they looked them up in a corpus of Facebook posts written by Poland travel agencies, in order to quantify their frequency of occurrence;

- they applied statistical correlation techniques to assess whether there is a correlation between the frequency distribution of PLTs and a number of variables, including the season (summer or winter) when the posts were written, the number of likes, comments, etc.;

- they assess whether the frequency distribution of "phrases containing linguistic tricks identical or semantically convergent to the post publicized by the travel agencies" corresponds to the frequency in a collection of comments to the corresponding posts.

Even if the main statements and goals of the paper are quite clear, I think that the paper has some flaws and needs to be revised in some parts before publication.

Let me start from the literature about the persuasive language processing and, more specifically, about the studies tackling the use of "linguistic tricks" for marketing communication purposes. At the end of the introduction, the authors claim: "Due to their specificity as well as narrow thematic scope, the issues of linguistic tricks and their usage in marketing communication in social media are rarely found in the scientific literature.". I suggest the authors to include in their survey of related work, the literature on Persuasive Language Processing, written in the Natural Language Processing community, such as for example the following titles:

>> Persuasive Language and Virality in Social Networks, Carlo Strapparava, Marco Guerini, Gözde Özbal (2011). In Proceedings of the International Conference on Affective Computing and Intelligent Interaction

>> Sentiment variations in text for persuasion technology, Lorenzo Gatti, Marco Guerini, Oliviero Stock, Carlo Strapparava (2014). In Proceedings of the International Conference on Persuasive Technology

>> Corps: A corpus of tagged political speeches for persuasive communication processing, Guerini Marco, Strapparava Carlo, Stock Oliviero (2008). In Journal of Information Technology and Politics

More specifically, I would suggest the following long journal paper for a good overview of what has been done for marketing communication purposes:

>> The Role of Syntax in Persuasive Marketing Communication: A Natural Language Processing Approach, Selin Atalay, Siham El Kihal, Florian Ellsaesser, 2020

Atalay, A. Selin and El Kihal, Siham and Ellsaesser, Florian, The Role of Syntax in Persuasive Marketing Communication: A Natural Language Processing Approach (January 10, 2020). Available at SSRN: https://ssrn.com/abstract=3410351 or http://dx.doi.org/10.2139/ssrn.3410351

Concerning the choice of the typologies of linguistic tricks, I would suggest the author to add some remarks and comments about the linguistic differences they found between the list they took from the literature, mostly in English, and the realization in the Polish posts. Some of the linguistic cues listed in Table 1 refer to specific English words, that, I guess, may have more than one translation in Polish.

Concerning Sections 4 and 5, I found the description of collected data and the interpretation of results not detailed enough. The authors start Section 4 by introducing the corpus of collected posts, but only in the following section we find out the internal composition of the corpus in terms of collected posts (i.e. a sub-corpus of 150 posts extracted from the wider corpus of 167). Since the frequency distribution of PLTs is computed for each travel agency, I would expected to see more details about the number of posts per agency. If the posts are not properly balanced across the five agencies, the uneven distribution may have an impact on the results reported in Table 2.

Still regarding Section 4: two of the main steps reported are not detailed sufficiently. The first one is described as follows: "the authors first noted the presence and the number of PLT in the content of each post in the spreadsheet". Since this kind of annotation task tends to be quite subjective, I would like to see an inter-annotator agreements score, if the task was carried out by the two authors. Instead, if the task was carried out only by one of the author, the annotation could be biased by the single annotator perception.

The second step is described as follows: "The content of the comments was then checked. The presence of identical or convergent semantic elements found in their content was also noted.". Also in this case, my concern is about the subjectivity of the phrase "identical or convergent semantic elements". How did you compute such semantic similarity? This computation issue, similarly to the aforementioned issue regarding the subjectivity of annotation, is mostly related to the future reproducibility of the data. Did the authors take into consideration this issue?

The authors claim that in order to assess the "impact of using emotive PLTs on the weight of the post" they used a specific metric whose formula is reported (Stepaniuk [8485]). Unfortunately, if I am not wrong, they did not report the results. Have I missed something? Still concerning the formula, they detailed all the component elements but they forgot to make explicit what the two constants, i.e. 4 and 16, refer to.

Minor issue regarding Section 4: I'm not familiar with STATISTICA, however I was wondering whether the authors chose to use version 12, instead of the last version.

Section 5 is quite hard to follow. The authors report in Table 2 the frequency distribution of PLTs in the different groups of posts they considered, but a specific table showing the different correlation values (together with their statistical significance) is missing. These values are reported and discussed in the paper, but I suggest to highlight them properly since they represent the main results of the study. As I have already written, I think that the absolute frequency of PLTs in the different groups of posts is not highly informative, since the groups contain a different total amount of posts.

This section should be the core of the paper, but unfortunately it contains very few hypothesis and more in-depth analysis of the obtained results. Even if the authors devoted the discussion to the following section, I suggest to add more specific remarks and comments about how the considered PLTs changes across posts.

The last part of this section is devoted to report the frequency of PLTs ("identical or semantically convergent to the post") in users' comments. I have already stated my concern about the subjectivity of considering a linguistic expression similar to another without any computational metric. In addition to this, I would also ask to the author how they isolated the "elements of content" that they considered in this last analysis. They carried out the whole analysis considering 72 elements of content for the summer post and 95 for the winter ones. Are these elements of content a subset of the whole corpus of posts they considered in the previous analyses? Why did they choose these elements? Lastly, the authors did not make the comments available, but only the posts of the travel agencies.

Minor issue regarding Section 5: when the authors write Table 1, I think that they would like to refer to Table 2.

The whole manuscript should be proof-read by a native speaker of English.

6. PLOS authors have the option to publish the peer review history of their article (what does this mean?). If published, this will include your full peer review and any attached files.

Reviewer #1: No

Reviewer #2: No

---

## [Author Response · Author response to Decision Letter 0]

16 Apr 2021

Dear Reviewers,

thank you for your time and engagement which, probably helps me to improve our manuscript. All of your remarks were relevant and were implemented in the current version of paper. Below, we want to briefly explain some of the improvements in our article according to your feedback. Our answers are marked in blue.

Reviewer 1:

Introduction

• A concise introduction is required to enable the reader’s understanding of the research problem.

• Expand to emphasize the problem leading to a clear set of research questions and objectives the research addresses.

Literature review

Use sub-headings to organize topics. Some critical studies are not included. The paper should relate coherently and convincingly with issues of real-world significance. This is a crucial phase contributing to research design. The theoretical framework emerging from the literature review could research questions and points of emphasis. 

Suggestions

• Consider summarizing the text based on the purpose of the study. – the purposes of the study was more clearly specified according to Reviewer sugesstion

• I wouldn’t recommend differentiating between the “main” aim and the “specific” aim as is presented in the manuscript, as it might be confusing to the reader. – such division into two types of goals has been removed

• Focus more on the empirical studies’ backgrounds. – the theoretical background of the paper was strengthened, new citations were added according to Reviewers sugestion

• Add more information to enable readers’ understanding of the authors’ view. - to achieve highier level of understanding by readers the article was reworded and checked by native speaker Editor

• You are encouraged to write concisely. The text can be reduced significantly.

Findings and discussion

Needs clear and comprehensive explanations to assist readers’ understanding. – a few issues were explained and described in a simpler way, and in order to achieve a higher level of understanding by readers, the article was checked by a native speaker

Limitations

There is no mention of the limitations of this study. – the limitations were added according to Reviewer remark

Summary

The study presented an important topic that would be of interest to the readership of this journal. Most research is relevant, including to the international audience. However, the research is missing a level of detail needed to understand the study result, the impact of the results, and the research contribution. Perhaps the authors are likely so close to the topic they are skipping over details that they know, but the reader would not. – a few issues were explained and described in a simpler way, and in order to achieve a higher level of understanding by readers, the article was checked by a native speaker

Overall, the paper requires more focus. The areas requiring attention are highlighted in the individual sections. In summary, the paper needs:

• A re-write of the abstract to give a good summary of the paper and mention the key concepts. - a few sentences from paper description by Reviewer 2 were used in new version of abstract. It was really helpful. Thus, the current version of abstract is more understandable.

• Expanding the introduction by clearly stating the research problem to suitably inform the reader. – the aims of the paper as well as its theoretical background were detailed

• A synthesized and structured critique of the literature. - done

• Clarifying the research procedures with an adequate explanation of the methods.

• Expanding the discussion to allow writing a well-developed conclusion summarizing the entire paper. The outcomes should be discussed in relation to the existing research. - done

• Emphasizing the significance of the research – a clear showing of how the findings contribute to new knowledge. – I think we did it when we wrote about evolutionary approach in the context of population genetics foundations in social media content analysis

• Using results to support the claims in conclusion adequately, and how the results of the research can be used for future research. - done

• The authors are encouraged to write concisely. The text can be reduced significantly. – part of the text was deleted

• In general, the language in the present manuscript is not of publication quality and requires improvement. Please carefully proof-read and spell-check to eliminate grammatical errors. – done (native speaker editor)

• The manuscript contains multiple stylistic, grammatical and conceptual flaws. The authors should make sure to differentiate between hyphens and dashes, look into the grammar more carefully (mistakes include “It consist” instead of “It consists”, “Sections 5” instead of “Section 5” and many others), revise intext references (at least one of them is included in the text in full). - done

Reviewer 2:

Let me start from the literature about the persuasive language processing and, more specifically, about the studies tackling the use of "linguistic tricks" for marketing communication purposes. At the end of the introduction, the authors claim: "Due to their specificity as well as narrow thematic scope, the issues of linguistic tricks and their usage in marketing communication in social media are rarely found in the scientific literature.". I suggest the authors to include in their survey of related work, the literature on Persuasive Language Processing, written in the Natural Language Processing community, such as for example the following titles:

>> Persuasive Language and Virality in Social Networks, Carlo Strapparava, Marco Guerini, Gözde Özbal (2011). In Proceedings of the International Conference on Affective Computing and Intelligent Interaction

>> Sentiment variations in text for persuasion technology, Lorenzo Gatti, Marco Guerini, Oliviero Stock, Carlo Strapparava (2014). In Proceedings of the International Conference on Persuasive Technology

>> Corps: A corpus of tagged political speeches for persuasive communication processing, Guerini Marco, Strapparava Carlo, Stock Oliviero (2008). In Journal of Information Technology and Politics

More specifically, I would suggest the following long journal paper for a good overview of what has been done for marketing communication purposes:

>> The Role of Syntax in Persuasive Marketing Communication: A Natural Language Processing Approach, Selin Atalay, Siham El Kihal, Florian Ellsaesser, 2020

Atalay, A. Selin and El Kihal, Siham and Ellsaesser, Florian, The Role of Syntax in Persuasive Marketing Communication: A Natural Language Processing Approach (January 10, 2020). Available at SSRN: https://ssrn.com/abstract=3410351 or http://dx.doi.org/10.2139/ssrn.3410351

- All the suggested literature sources are cited in the article.

Concerning the choice of the typologies of linguistic tricks, I would suggest the author to add some remarks and comments about the linguistic differences they found between the list they took from the literature, mostly in English, and the realization in the Polish posts. Some of the linguistic cues listed in Table 1 refer to specific English words, that, I guess, may have more than one translation in Polish. – Thank you for your remark. Most of literature resources cited in our typology have their translations and editions in Polish, and the examples of persuasive communication come from there (i.a Hogan and Speakman, 2006; Cialdini, 2008; Sutherland and Sylvester, 2000; Hogan, 2010). Almost all PLT examples can be based on these items. Moreover, examples of persuasive communication have their direct equivalents in Polish. The purpose of referring our typology to only English-language scientific resources was to make it universal and more diverse from the perspective of scientific sources. It was a suggestion from our internal reviewers. Perhaps this was a wrong assumption.

Concerning Sections 4 and 5, I found the description of collected data and the interpretation of results not detailed enough. The authors start Section 4 by introducing the corpus of collected posts, but only in the following section we find out the internal composition of the corpus in terms of collected posts (i.e. a sub-corpus of 150 posts extracted from the wider corpus of 167). Since the frequency distribution of PLTs is computed for each travel agency, I would expected to see more details about the number of posts per agency. If the posts are not properly balanced across the five agencies, the uneven distribution may have an impact on the results reported in Table 2. – the data concerning the frequency of posts per travel offices were included in Results section

Still regarding Section 4: two of the main steps reported are not detailed sufficiently. The first one is described as follows: "the authors first noted the presence and the number of PLT in the content of each post in the spreadsheet". Since this kind of annotation task tends to be quite subjective, I would like to see an inter-annotator agreements score, if the task was carried out by the two authors. Instead, if the task was carried out only by one of the author, the annotation could be biased by the single annotator perception.

The second step is described as follows: "The content of the comments was then checked. The presence of identical or convergent semantic elements found in their content was also noted.". Also in this case, my concern is about the subjectivity of the phrase "identical or convergent semantic elements". How did you compute such semantic similarity? This computation issue, similarly to the aforementioned issue regarding the subjectivity of annotation, is mostly related to the future reproducibility of the data. Did the authors take into consideration this issue? – the data concerning the frequency of posts per travel offices were included in Results section. The phrases concerning „identical or convergent semantic elements” were incorrect and were deleted according to Reviewer remark.

The authors claim that in order to assess the "impact of using emotive PLTs on the weight of the post" they used a specific metric whose formula is reported (Stepaniuk [8485]). Unfortunately, if I am not wrong, they did not report the results. Have I missed something? Still concerning the formula, they detailed all the component elements but they forgot to make explicit what the two constants, i.e. 4 and 16, refer to. – these analyses were present in the subsection "Seasonality, frequency of emotional PLT and post range. " The visibility of aforementioned results were highlited inthe methodology section as well as in the results section.

Minor issue regarding Section 4: I'm not familiar with STATISTICA, however I was wondering whether the authors chose to use version 12, instead of the last version. – information about the newest version of Statistica was added

Section 5 is quite hard to follow. The authors report in Table 2 the frequency distribution of PLTs in the different groups of posts they considered, but a specific table showing the different correlation values (together with their statistical significance) is missing. These values are reported and discussed in the paper, but I suggest to highlight them properly since they represent the main results of the study. As I have already written, I think that the absolute frequency of PLTs in the different groups of posts is not highly informative, since the groups contain a different total amount of posts.

This section should be the core of the paper, but unfortunately it contains very few hypothesis and more in-depth analysis of the obtained results. Even if the authors devoted the discussion to the following section, I suggest to add more specific remarks and comments about how the considered PLTs changes across posts.

The last part of this section is devoted to report the frequency of PLTs ("identical or semantically convergent to the post") in users' comments. I have already stated my concern about the subjectivity of considering a linguistic expression similar to another without any computational metric. In addition to this, I would also ask to the author how they isolated the "elements of content" that they considered in this last analysis. They carried out the whole analysis considering 72 elements of content for the summer post and 95 for the winter ones. Are these elements of content a subset of the whole corpus of posts they considered in the previous analyses? Why did they choose these elements? Lastly, the authors did not make the comments available, but only the posts of the travel agencies. – Thank you for your remark. Of course, in this case we agree with the Reviewer. When collecting the data, we considered that comments are an integral part of each post, so having access to the content of the posts and other attributes enabling their identification, we did not see the need to archive them. The analyzes were conducted using our typology. Only the presence of a given element was marked in the post text corpus and in the comment. Then the totals were entered into the worksheet. At that time, we were more concerned with quantitative relationships, the qualitative relationships were skipped. This is probably our mistake. But we want to go back to such analysis next time.

The tables with relevant data were added according to Reviewer remark.

Minor issue regarding Section 5: when the authors write Table 1, I think that they would like to refer to Table 2. – done 

The whole manuscript should be proof-read by a native speaker of English. – done 

Thank you, 

Authors

---

## [Decision Letter · Decision Letter 1]

2 Jun 2021

PONE-D-21-02199R1

Persuasive Linguistic Tricks in Social Media Marketing Communication - the Memetic Approach

PLOS ONE

Dear Dr. Stepaniuk,

Thank you for submitting your manuscript to PLOS ONE. After careful consideration, we feel that it has merit but does not fully meet PLOS ONE’s publication criteria as it currently stands. Therefore, we invite you to submit a revised version of the manuscript that addresses the points raised during the review process.

Specifically, while the manuscript has clearly improved in the previous round of revision, a few minor edits are still needed, as thoroughly specified by Reviewer #2. Please, address the remaining issues before resubmitting. I expect the next round of review to be rather quick, as I will not send the manuscript our for revision but I will personally check your edits as soon as I'll receive the revised paper.

We look forward to receiving your revised manuscript.

Kind regards,

Stefano Cresci

Academic Editor

PLOS ONE

Journal Requirements:

Additional Editor Comments (if provided):

Both reviewers agree that the authors have addressed the majority of their concerns. As such, the manuscript has significantly improved.

Only a few minor adjustments are still needed before the manuscript can be granted publication. Specifically, authors should tone down their claim in the introduction that persuasive linguistic tricks are rarely discussed in literature. Perhaps the most important edit that authors are asked to implement, is related to the limitations of their work. They should clearly specify the possibility for some missing Polish phrases, due to their choice of working with the English language. Moreover, they should also clearly mention and discuss the low correlations that they obtained, and its implications. Finally, they should also better clarify and explain what they mean by "elements of content", as Reviewer #2 noted, and they should check the consistency of their p-values throughout the manuscript.

Reviewers' comments:

Reviewer's Responses to Questions

**Comments to the Author**

1. If the authors have adequately addressed your comments raised in a previous round of review and you feel that this manuscript is now acceptable for publication, you may indicate that here to bypass the “Comments to the Author” section, enter your conflict of interest statement in the “Confidential to Editor” section, and submit your "Accept" recommendation.

Reviewer #1: All comments have been addressed

Reviewer #2: (No Response)

2. Is the manuscript technically sound, and do the data support the conclusions?

Reviewer #1: Yes

Reviewer #2: Yes

3. Has the statistical analysis been performed appropriately and rigorously? 

Reviewer #1: Yes

Reviewer #2: Yes

4. Have the authors made all data underlying the findings in their manuscript fully available?

Reviewer #1: Yes

Reviewer #2: Yes

5. Is the manuscript presented in an intelligible fashion and written in standard English?

Reviewer #1: Yes

Reviewer #2: Yes

6. Review Comments to the Author

Reviewer #2: Concerning my first remark about missing literature, the authors included in the paper the works I suggested. However, in the introduction they still claim that "the issues of persuasive linguistic tricks and their use in social media marketing communication are rarely found in the scientific literature". My personal opinion is that this claim is not true.

Second remark about the choice of the typologies of linguistic tricks uniquely based on English literature. Even if I'm not a Polish speaker, of course I agree that the English phrases have an equivalent in Polish. Still, I'm quite sure that there are many other native-Polish phrases that you are missing in your research since it is grounded on English phrases.

I appreciate that the authors took into consideration my remarks about the description of results. However, I still do not find where the authors illustrate what the "elements of content" are. Unless, the authors refer to the linguistic phrases as "elements of content". I think that this issue should be clarified before the publication.

I think that the quality of the paper was generally improved after the first round of revisions. However, I think that some minor revisions are needed before publication. The authors should highlighted that they obtained very low correlations, considering all the variables. This should be more stressed. I'm not claiming that it is not a valuable result, but I think it should discussed more in detailed.

Minor remarks: check carefully the direction of the p-value, it is not consistent through the paper. It should be < but in some cases you wrote >.

7. PLOS authors have the option to publish the peer review history of their article (what does this mean?). If published, this will include your full peer review and any attached files.

Reviewer #1: No

Reviewer #2: No

---

## [Author Response · Author response to Decision Letter 1]

4 Jun 2021

Dear Reviewers,

thank you for your time and engagement which, probably helps me to improve our manuscript. All of your remarks were relevant and were implemented in the current version of paper. Below, we want to briefly explain some of the improvements in our article according to your feedback. Our answers are marked in blue.

Reviewer #2: Concerning my first remark about missing literature, the authors included in the paper the works I suggested. However, in the introduction they still claim that "the issues of persuasive linguistic tricks and their use in social media marketing communication are rarely found in the scientific literature". My personal opinion is that this claim is not true. 

Thank you for this remark. Indeed, that statement went too far. The present version emphasizes the presence in the article of a new, interdisciplinary and evolutionary approach to the issues of linguistic persuasion in SNS.

Second remark about the choice of the typologies of linguistic tricks uniquely based on English literature. Even if I'm not a Polish speaker, of course I agree that the English phrases have an equivalent in Polish. Still, I'm quite sure that there are many other native-Polish phrases that you are missing in your research since it is grounded on English phrases. 

Thank you for this remark. The issue of adapting the typology used to the specificity of national languages has been emphasized in the " Limitations of the Study and Future Research" section.

I appreciate that the authors took into consideration my remarks about the description of results. However, I still do not find where the authors illustrate what the "elements of content" are. Unless, the authors refer to the linguistic phrases as "elements of content". I think that this issue should be clarified before the publication. 

The phrase "element of content" was used interchangeably by the authors with a "Facebook post". It has been removed and replaced with a simple "post(s)".

I think that the quality of the paper was generally improved after the first round of revisions. However, I think that some minor revisions are needed before publication. The authors should highlighted that they obtained very low correlations, considering all the variables. This should be more stressed. I'm not claiming that it is not a valuable result, but I think it should discussed more in detailed. 

Thank you. The issue of low values of correlation coefficients was additionally discussed in „Discussion” section.

Minor remarks: check carefully the direction of the p-value, it is not consistent through the paper. It should be < but in some cases you wrote >. 

Those errors have been fixed.

Thank you, 

Authors

---

## [Editor Report · Decision Letter 2]

17 Jun 2021

Persuasive Linguistic Tricks in Social Media Marketing Communication - the Memetic Approach

PONE-D-21-02199R2

Dear Dr. Stepaniuk,

We’re pleased to inform you that your manuscript has been judged scientifically suitable for publication and will be formally accepted for publication once it meets all outstanding technical requirements.

Kind regards,

Stefano Cresci

Academic Editor

PLOS ONE

---

## [Editor Report · Acceptance letter]

23 Jun 2021

PONE-D-21-02199R2 

Persuasive Linguistic Tricks in Social Media Marketing Communication — the Memetic Approach 

Dear Dr. Stepaniuk:

I'm pleased to inform you that your manuscript has been deemed suitable for publication in PLOS ONE. Congratulations! Your manuscript is now with our production department. 

Kind regards, 

on behalf of

Dr. Stefano Cresci 

Academic Editor

PLOS ONE